# Population-scale analysis of common and rare genetic variation associated with hearing loss in adults

Kavita Praveen[1✉], Lee Dobbyn[1], Lauren Gurski[1], Ariane H. Ayer[1], Jeffrey Staples[1], Shawn Mishra[2], Yu Bai [2], Alexandra Kaufman[2], Arden Moscati[1], Christian Benner[1], Esteban Chen[1], Siying Chen[1], Alexander Popov[1], Janell Smith[2], GHS-REGN DiscovEHR collaboration*, Regeneron Genetics Center*, Decibel-REGN collaboration*, Olle Melander[3,4], Marcus B. Jones[1], Jonathan Marchini [1], Suganthi Balasubramanian[1], Brian Zambrowicz [2], Meghan C. Drummond[2], Aris Baras [1], Goncalo R. Abecasis [1], Manuel A. Ferreira [1], Eli A. Stahl [1] & Giovanni Coppola [1✉]

To better understand the genetics of hearing loss, we performed a genome-wide association meta-analysis with 125,749 cases and 469,497 controls across five cohorts. We identified 53/c loci affecting hearing loss risk, including common coding variants in *COL9A3* and *TMPRSS3*. Through exome sequencing of 108,415 cases and 329,581 controls, we observed rare coding associations with 11 Mendelian hearing loss genes, including additive effects in known hearing loss genes *GJB2* (Gly12fs; odds ratio [OR] = 1.21, $P = 4.2 \times 10^{-11}$) and *SLC26A5* (gene burden; OR = 1.96, $P = 2.8 \times 10^{-17}$). We also identified hearing loss associations with rare coding variants in *FSCN2* (OR = 1.14, $P = 1.9 \times 10^{-15}$) and *KLHDC7B* (OR = 2.14, $P = 5.2 \times 10^{-30}$). Our results suggest a shared etiology between Mendelian and common hearing loss in adults. This work illustrates the potential of large-scale exome sequencing to elucidate the genetic architecture of common disorders where both common and rare variation contribute to risk.

[1] Regeneron Genetics Center, Tarrytown, NY 10591, USA. [2] Regeneron Pharmaceuticals, Inc., Tarrytown, NY 10591, USA. [3] Lund University, Department of Clinical Sciences Malmö, Malmö, Sweden. [4] Skåne University Hospital, Department of Emergency and Internal Medicine, Malmö, Sweden. *Lists of authors and their affiliations appear at the end of the paper. ✉email: kavita.praveen@regeneron.com; giovanni.coppola@regeneron.com

The loss of hearing can have a debilitating impact on quality of life, requiring major adjustments to day-to-day activities. Serious comorbidities are also associated with hearing loss including social isolation, depression, cognitive impairment, and dementia, which further deteriorate quality of life[1]. Disabling hearing loss is common, with ~466 million people affected worldwide (World Health Organization: https://www.who.int/health-topics/hearing-loss). While existing technologies—such as hearing aids and cochlear implants—can ameliorate hearing loss, their use is limited by barriers to access, including cost, health policies and regulations, and social stigma associated with device use[2]. Furthermore, while these assistive devices typically provide some benefit, they do not address the chief complaint associated with acquired hearing loss: lack of hearing clarity, particularly in social and work environments (https://www.hearingloss.org/programs-events/patient-focused-drug-development-meeting). The Lancet Commission on dementia prevention, intervention, and care has identified untreated hearing loss in middle age as the top modifiable risk factor for dementia, but it is estimated that 67–86% of adults who may benefit from hearing aids do not use them[1,3]. These challenges, combined with a lack of therapeutics to stop or slow hearing loss progression, have contributed to its status as a growing global health issue. Novel therapies based on genetic evidence, therefore, will be crucial in addressing this unmet need.

Hearing loss affects individuals of all ages, but its prevalence increases with age. Approximately 1-2/1000 babies are born with hearing loss[4]. Mutations in over 150 genes (https://hereditaryhearingloss.com) account for over 50% of the cases. While autosomal recessive hearing loss is generally pre-lingual and non-progressive, autosomal dominant forms are mostly post-lingual (including adult onset) and progressive. The prevalence of hearing loss increases to 2.8/1,000 in primary school-age children and 3.5/1000 in adolescents[4,5]. The National Institute on Deafness and other Communication Disorders calculates that by the age of 45 ~2% of individuals have a disabling hearing loss, and this number increases to 50% in individuals over the age of 75. This increase in prevalence with age reflects a combination of late-onset hearing loss mutations, the cumulative effects of environmental factors such as exposure to noise and ototoxic drugs and, in aging individuals, the degenerative effects of age on the cochlea. These genetic and environmental insults primarily damage the structures of the inner ear, resulting in sensorineural hearing loss[6,7].

Heritability estimates for age-related hearing loss range as high as 36 and 70%[8–12] suggesting that genetics, along with environmental factors, play a major role in determining an individual's risk for developing hearing loss. Genome-wide association studies (GWAS) of hearing loss in adults have identified 61 common variant loci associated with the trait in Europeans[13–20]. Three of these studies[18–20] have included genotyping and imputed data from UK Biobank as well. While the majority of these studies have established a common variant contribution to adult hearing loss, few reports have addressed the contibution of rare and low-frequency variation[21].

Recently, Ivarsdottir et al.[18] published association results with hearing loss on ~50,000 Icelandic individuals with whole-genome sequence and ~50,000 individuals from UKB with exome sequence data, with imputation of larger samples into these variant sets. We have now expanded the rare variant analysis to exome-sequences from ~294,710 individuals in UKB and ~143,286 individuals from three other datasets. Here, we report findings from genome- and exome-wide association meta-analyses including 125,749 cases and 469,497 controls. Our analyses have identified 15 susceptibility loci that were previously not associated with hearing loss, to the best of our knowledge, and 15 rare variant associations that provide important insights into the biology of hearing loss in adults.

## Results

To study common variants, we performed association meta-analyses using genotyping and imputation across five cohorts: the Geisinger DiscovEHR study (GHS), the Malmö Diet and Cancer study from Malmö, Sweden (MALMO), the Mount Sinai's Bio*Me* Personalized Medicine Cohort from Mount Sinai Health System, New York (SINAI), UK Biobank (UKB) and an additional study from Finland, FinnGen, for a total of 125,749 cases and 469,497 controls. To study rare and ultra-rare variants, we also generated exome sequence data and performed combined GWAS and exome-wide association study (ExWAS) on a subset of 108,415 cases and 329,581 controls across GHS, MALMO, SINAI, and UKB. Phenotypes were derived from ICD-10 diagnosis codes in GHS, MALMO, SINAI, and FinnGen, and combined self-report and ICD-10 codes in UKB (see Methods and Supplementary Data 1 for details). Our genome-wide association analyses included 15,881,489 variants with frequency > 0.1% that were genotyped or imputed with $r^2 > 0.3$ in at least one study, and 2,923,124 coding or essential splice site variants with minor allele count at least five from exome sequencing (111,588 of which overlapped with the imputed).

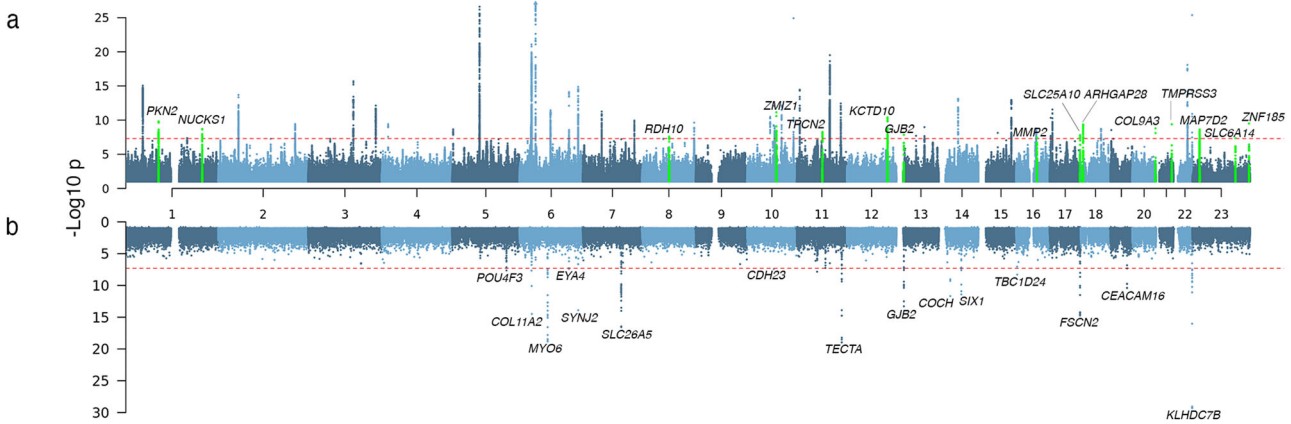

**Fig. 1 Associations from the meta-analysis of hearing loss across 5 cohorts. a** Common (MAF ≥ 0.01) variant associations with hearing loss. Colored in green are loci that have not previously been associated with hearing loss. **b** Rare (MAF < 0.01) coding single-variant and gene burden associations with hearing loss. The gene labels refer to the nearest gene.

**Table 1 Lead variants at the 15 loci not previously reported to be associated with hearing loss in GWAS.**

| Position | rsID | OR (LCI, UCI) | p-value | AAF | nSNPs | Nearest Gene | Direction |
|----------|------|---------------|---------|-----|-------|--------------|-----------|
| 10:78760556:T:C | rs11596052 | 0.959 (0.947, 0.970) | 7.14E−12 | 0.2162 | 94 | ZMIZ1 | +−−−− |
| 12:109460927:A:G | rs1558804 | 1.033 (1.023, 1.044) | 4.11E−11 | 0.4468 | 620 | KCTD10 | −++++ |
| 1:88282464:G:A | rs475788 | 1.036 (1.024, 1.047) | 1.56E−10 | 0.7145 | 462 | PKN2 | −++++ |
| 23:152898837:T:C | rs186256023 | 1.082 (1.056, 1.109) | 3.14E−10 | 0.0315 | 16 | ZNF185 | +++−? |
| 21:42388983:C:T | rs45598239 | 1.070 (1.048, 1.093) | 4.05E−10 | 0.0549 | 18 | TMPRSS3 | +++++ |
| 18:6678716:T:A | rs8090563 | 1.031 (1.021, 1.042) | 4.29E−10 | 0.4901 | 210 | ARHGAP28 | +++++ |
| 20:62819980:C:T | rs61734651 | 1.062 (1.041, 1.083) | 1.60E−09 | 0.0654 | 57 | COL9A3 | +++++ |
| 1:205751355:G:A | rs823116 | 0.971 (0.961, 0.980) | 1.99E−09 | 0.5493 | 209 | NUCKS1 | +−−−− |
| 23:20054286:C:T | rs7055595 | 1.033 (1.022, 1.044) | 2.44E−09 | 0.2007 | 488 | MAP7D2 | ++++? |
| 11:69208057:T:C | rs7926098 | 1.030 (1.020, 1.041) | 5.43E−09 | 0.6182 | 204 | TPCN2 | −++−+ |
| 13:20186225:A:T | rs117887149 | 1.114 (1.073, 1.156) | 1.40E−08 | 0.0178 | 10 | GJB2 | +++−+ |
| 17:81711135:C:T | rs62077192 | 1.140 (1.089, 1.193) | 1.55E−08 | 0.0117 | 92 | SLC25A10 | +++−+ |
| 8:73332501:T:C | rs4738323 | 1.035 (1.023, 1.048) | 2.59E−08 | 0.1972 | 42 | RDH10 | −++++ |
| 16:55456852:T:A | rs17300627 | 0.967 (0.955, 0.978) | 3.88E−08 | 0.2034 | 105 | MMP2 | +−−−− |
| 23:116435671:G:A | rs3788766 | 1.025 (1.016, 1.034) | 4.35E−08 | 0.6386 | 66 | SLC6A14 | +++−? |

The alternate allele frequency (AAF) refers to the allele listed second in the 'Position' column. The nSNPs column indicates the number of genome-wide significant SNPs within each locus. The direction of the effect in each study (in the order: MALMO, UKB, GHS, SINAI, FinnGen) in the meta-analysis is given in the 'Direction' column, where '+' indicates increased risk, '−' indicates decreased risk, and '?' indicates that the variant was not present or tested.

**Common variant associations**. We identified 53 independent loci harboring genome-wide significant ($P < 5 \times 10^{-8}$) common (MAF ≥ 0.01) variants associated with hearing loss (Fig. 1a, Supplementary Data 2, Supplementary Fig. 1), 38 of which are shared with the previously reported 61 hearing loss-associated loci. We observed a genomic control lambda_GC = 1.36 but an LD score regression intercept 1.054 (standard error 0.008), indicating that inflation was largely due to polygenic signal for the hearing loss phenotype.

Among the lead variants at the 15 loci uncovered in this analysis (Fig. 1a, Table 1, Supplementary Fig. 2) is rs117887149 that maps close to *GJB2* (Supplementary Fig. 2g), which is a predominant cause of congenital hearing loss. While the majority of lead variants in these loci lay in intergenic or downstream/upstream regions of genes, at two loci they were within the introns of the following genes: *KCTD10*, a member of the potassium channel tetramerization domain family that is implicated in cardiac development[22,23] and *MAP7D2*, an axonal cargo transport protein predominantly expressed in the brain[24].

Top variants at two other loci were missense changes and thus also implicate specific genes: in *COL9A3* (Arg103Trp; MAF = 0.07) and *TMPRSS3* (Ala90Thr; MAF = 0.06). Mutations in *COL9A3*, which is highly expressed in the ear, have been implicated in the autosomal recessive Stickler syndrome, in which hearing loss is prominent[25], and tentatively in non-syndromic hearing loss[26,27]. High-throughput mouse knockout characterization from the International Mouse Phenotyping Consortium (IMPC; https://www.mousephenotype.org) indicates that *Col9a3*-null mice also have hearing loss[28]. *TMPRSS3* is a type II membrane serine protease that localizes to the endoplasmic reticulum and plasma membranes, and is expressed in hair cells and supporting cells in the organ of Corti, the spiral ganglion, and the stria vascularis in the ear[29–31]. Mutations in *TMPRSS3* cause congenital and childhood-onset autosomal recessive hearing loss[32] but there is also evidence for hearing deficits in heterozygous carriers[18].

The number of associated variants at the 50 autosomal loci ranged from eight to 1326. In order to assess the presence of multiple independent causal variants at each locus, we ran conditional analyses using GCTA-COJO[33], which indicated the presence of secondary association signals (joint $P < 10^{-5}$) at eight loci (Supplementary Data 3). We also ran FINEMAP Bayesian causal variant inference[34] to prioritize associated variants as credibly causal at thirty loci that were genome-wide significant in

RGC data (excluding FinnGen). The COJO and FINEMAP methods disagreed on the presence of single vs. multiple association signals within five loci, three of which showed subthreshold ($10^{-5} < P < 10^{-4}$) secondary associations in COJO. FINEMAP prioritized ten or fewer variants in top causal variant 95% credible sets for the top causal variant at eleven loci, including single putatively causal variants at four loci (Supplementary Data 3): missense variants in *CDH23* (Ala371Thr; rs143282422) and *KLHDC7B* (Val504Met; rs36062310), and intronic variants in *CTBP2* (rs10901863) and *PAFAH1B1* (rs12938775).

**Colocalization with GTEx eQTLs identifies candidate genes driving GWAS signals**. To identify genes for which expression regulation might drive the observed association signals, we tested for colocalization of our hearing loss-associated loci with expression quantitative trait loci (eQTL) data for 48 tissues from the Genotype-Tissue Expression (GTEx; https://www.gtexportal.org) project using coloc2[35]. Across all GTEx eQTL tissues tested, we identified 19 genes mapping to 15 loci with evidence (posterior probability of colocalization, PPH4 ≥ 0.5) for colocalization between the hearing loss association and an eQTL signal, in at least one tissue (Supplementary Data 4). Only two of the 15 loci with GWAS-eQTL overlap had multiple eQTL signals for more than one gene (*NUCKS1* and *RAB29* in locus 1-4; *ACADVL*, *DLG4*, *CTDNEP1*, and *CLDN7* in locus 17-2) making it difficult to prioritize causal genes at these loci based on eQTL data. Since GTEx did not include tissues from the ear, we used single-cell RNA sequencing data that were generated in-house from mouse cochleae to check the expression of the 19 genes in the ear (Supplementary Data 5). Thirteen of the 19 genes showed evidence of expression across 26 inner-ear cell types and, of these, 12 were expressed in the hair cells. While the majority of genes showed broad expression across the 26 cell types, we noted a subset that were specific to only a few, including *CRIP3* in inner and outer hair cells, and *TCF19* in neurons and immune cells (Supplementary Data 4 & 5).

**Rare-variant association analysis identifies large-effect hearing loss variants in known hearing loss genes**. We identified significant ($P < 5 \times 10^{-8}$) rare variant (MAF < 0.01) associations in 25 genes, 15 of which had nonsynonymous variant or gene burden associations (Fig. 1b, Table 2, Supplementary Data 6 & 7).

**Table 2 Nonsynonymous rare (minor allele frequency, MAF < 0.01) variants and gene burdens associated ($P < 5 \times 10^{-8}$) with hearing loss in meta-analysis.**

| Gene | Top Burden/SNV | OR (LCI, UCI) | p-value | Direction |
|---|---|---|---|---|
| KLHDC7B | pLOF only (MAF ≤ 0.01) | 2.145 (1.881, 2.446) | 5.22E−30 | −+++? |
| TECTA | pLOF + strict deleterious missense (MAF ≤ 0.01) | 1.358 (1.271, 1.451) | 1.10E−19 | ++++? |
| MYO6 | pLOF + strict deleterious missense (MAF ≤ 0.0001) | 1.565 (1.420, 1.725) | 1.51E−19 | +++?? |
| FSCN2 | pLOF + all missense (MAF ≤ 0.01) | 1.144 (1.107, 1.183) | 1.90E−15 | +++−? |
| COL11A2 | 6:33189182:A:G; Phe80Ser | 6.926 (4.280, 11.208) | 3.24E−15 | ?+??? |
| SYNJ2 | 6:158071628:C:T; Thr656Met | 1.306 (1.221, 1.398) | 1.25E−14 | +++++ |
| SLC26A5 | pLOF and strict deleterious missense (MAF ≤ 0.0001) | 1.956 (1.674, 2.284) | 2.75E−17 | −++?? |
| COCH | pLOF + strict deleterious missense (MAF ≤ 0.0001) | 1.719 (1.449, 2.039) | 5.22E−10 | −++?? |
| SIX1 | pLOF + strict deleterious missense (MAF ≤ 0.00001) | 4.252 (2.825, 6.400) | 3.95E−12 | ?++?? |
| CEACAM16 | pLOF + deleterious missense (MAF ≤ 0.01) | 1.187 (1.128, 1.249) | 4.01E−11 | −++−? |
| GJB2 | 13:20189546:AC:A; Gly12fs | 1.214 (1.146, 1.286) | 4.23E−11 | −+++? |
| TBC1D24 | 16:2497068:A:G; Asn307Ser | 4.140 (2.571, 6.668) | 5.10E−09 | ?++?? |
| POU4F3 | pLOF + all missense (MAF ≤ 0.00001) | 1.923 (1.529, 2.418) | 2.31E−08 | ?++?? |
| CDH23 | 10:71712737:A:G; Asn1103Ser | 1.204 (1.127, 1.287) | 3.70E−08 | ++++? |
| EYA4 | pLOF only (MAF ≤ 0.001) | 3.077 (2.061, 4.595) | 3.87E−08 | +++?? |

The direction of the effect in each single study (in the order: MALMO, UKB, GHS, SINAI, FinnGen) in the meta-analysis is given in the 'Direction' column where '+' indicates increased risk, '−' indicates decreased risk and '?' indicates that the variant was not present or tested.

Five of the 15 genes also had significant common variant associations within 1 Mb (*KLHDC7B, SYNJ2, GJB2, EYA4*, and *CDH23*). After conditioning on independent common variants at each locus (Supplementary Data 6, 7 & 8), the rare variant and gene burdens remained associated with hearing loss (maximum conditional $P \le 3 \times 10^{-5}$) except for the *CDH23* Asn1103Ser association (conditional $P = 0.79$). Of note, the lead common variant in *CDH23* is also a missense (Ala371Thr) variant (MAF = 0.01) and is pinpointed by FINEMAP as the only causal variant at that locus with high confidence. Overall, our conditional analyses suggest that rare variant association signals are usually independent of nearby common variant associations.

*Associations with Mendelian hearing loss genes.* Of the 14 genes with independent, rare nonsynonymous and/or burden associations, 11 were previously identified as causes of Mendelian forms of hearing loss. These include associations with two genes (*GJB2* and *SLC26A5*) that cause recessive hearing loss, burden associations in seven genes (*MYO6, COCH, TECTA, SIX1, CEACAM16, POU4F3,* and *EYA4*) and single-variant associations in two genes (*TBC1D24* and *COL11A2*) that cause autosomal dominant hearing loss (reviewed in Shearer et al.[36]). In *MYO6* and *COCH*, we also observed genome-wide significant single-variant associations of large-effect size (*MYO6* His246Arg, OR = 30.7; *COCH* Cys542Phe, OR = 81.4; Supplementary Data 6). Both variants have previously been characterized as pathogenic in family-based genetic analyses[37,38]. Only a minority of variants included in burden tests have been classified as pathogenic by ClinVar (Supplementary Data 9) suggesting that our analysis has detected additional, risk-associated variants with variable penetrance in Mendelian hearing loss genes. We also identified single-variant associations in two genes that cause autosomal dominant hearing loss: *TBC1D24* Asn307Ser, also recently reported by Ivarsdottir et al.[18], was implicated as pathogenic in two unrelated families[39], and *COL11A2* Phe80Ser, which has not yet been classified as

pathogenic but is predicted deleterious, lies in a domain (Laminin G-like/NC4) that harbors other mutations causing non-syndromic hearing loss[40].

We performed our analysis under an additive model, which assumes risk effects in heterozygous carriers as well as homozygotes; therefore, it is not surprising to see that the majority of Mendelian genes (9/11) identified can cause hearing loss in heterozygote carriers. However, we also detect associations in two genes that have previously been implicated in recessive hearing loss: *GJB2* Gly12fs (OR = 1.21; $P = 4 \times 10^{-11}$), and *SLC26A5* Leu46Pro (OR = 1.3; $P = 3 \times 10^{-14}$) as well as the burden of *SLC26A5* predicted loss-of-function (pLOF) and strict deleterious missense variants (excluding Leu46Pro) (OR = 1.96; $P = 3 \times 10^{-17}$) (Fig. 2). We also observed a suggestive association with *GJB2* Leu90Pro (OR = 1.51, $P = 4.4 \times 10^{-5}$), another known pathogenic variant in recessive hearing loss. The associations persisted after excluding homozygous carriers and compound heterozygous carriers of rare, coding variants in these genes (Supplementary Data 10), suggesting a previously unappreciated increase in risk for hearing loss in heterozygous carriers of loss-of-function *GJB2*, and missense and pLOF *SLC26A5* variants.

*Rare-variant associations in KLHDC7B, FSCN2, and SYNJ2.* The most significant rare coding association in our analysis was the aggregate of 68 pLOF variants (43 frameshift, 23 stop-gain and 2 stop-loss, (Supplementary Data 9)) in *KLHDC7B* (Kelch-like domain containing 7B), with an approximately two-fold increase in risk for hearing loss (OR = 2.14, $P = 5 \times 10^{-30}$). The main contributors to the gene burden were two frameshift variants, Gly302fs and Lys181fs, that are predicted to truncate the protein near the start of the Kelch domains. These variants were also significantly associated in single-variant tests (Fig. 3), and Gly302fs was recently reported in an analysis that included UKB[18]. The association with pLOF variants remained significant after repeating the pLOF burden test conditioning on Gly302fs

#### a. GJB2 (Gly12fs)

| Study | Cases | Controls | MAF | OR (LCI \| UCI) | p-value |
|---|---|---|---|---|---|
| GHS | 9,296 \| 193 \| 4 | 97,381 \| 2,012 \| 1 | 0.0105 | 1.10 (0.94 \| 1.29) | 2.5E–01 |
| UKB | 104,059 \| 1,760 \| 15 | 207,696 \| 2,944 \| 0 | 0.0075 | 1.24 (1.16 \| 1.32) | 1.6E–11 |
| SINAI | 307 \| 7 \| 0 | 9,857 \| 163 \| 0 | 0.0082 | 1.38 (0.57 \| 3.35) | 4.8E–01 |
| MALMO | 497 \| 3 \| 0 | 26,707 \| 386 \| 0 | 0.0070 | 0.54 (0.26 \| 1.14) | 1.0E–01 |
| **Meta** | **114,159 \| 1,963 \| 19** | **341,641 \| 5,505 \| 1** | **0.0081** | **1.21 (1.15 \| 1.29)** | **4.2E–11** |

#### b. SLC26A5 (Leu46Pro)

| Study | Cases | Controls | MAF | OR (LCI \| UCI) | p-value |
|---|---|---|---|---|---|
| GHS | 8,935 \| 92 \| 0 | 95,478 \| 831 \| 2 | 0.0044 | 1.15 (0.91 \| 1.45) | 2.5E–01 |
| UKB | 104,469 \| 1,359 \| 6 | 208,576 \| 2,052 \| 12 | 0.0054 | 1.32 (1.23 \| 1.42) | 2.2E–14 |
| SINAI | 312 \| 2 \| 0 | 9,940 \| 82 \| 0 | 0.0041 | 0.97 (0.24 \| 4.00) | 9.7E–01 |
| MALMO | 493 \| 7 \| 0 | 26,761 \| 349 \| 2 | 0.0063 | 1.19 (0.51 \| 2.79) | 6.9E–01 |
| FinnGen | 9,500 \| 106 \| 0 | 121,070 \| 1,253 \| 3 | 0.0016 | 1.09 (0.75 \| 1.58) | 6.6E–01 |
| **Meta** | **123,709 \| 1,566 \| 6** | **461,825 \| 4,567 \| 19** | **0.0052** | **1.30 (1.21 \| 1.39)** | **3.1E–14** |

#### c. SLC26A5 aggregate of pLOF and strict deleterious missense

| Study | Cases | Controls | MAF | OR (LCI \| UCI) | p-value |
|---|---|---|---|---|---|
| GHS | 9,002 \| 25 \| 0 | 96,103 \| 208 \| 0 | 0.0011 | 1.51 (0.92 \| 2.47) | 1.0E–01 |
| UKB | 98,267 \| 307 \| 0 | 195,806 \| 330 \| 0 | 0.0011 | 2.03 (1.73 \| 2.40) | 2.5E–17 |
| MALMO | 500 \| 0 \| 0 | 27,056 \| 56 \| 0 | 0.0010 | NA | 3.4E–01 |
| **Meta** | **107,769 \| 332 \| 0** | **318,965 \| 594 \| 0** | **0.0011** | **1.96 (1.67 \| 2.28)** | **2.8E–17** |

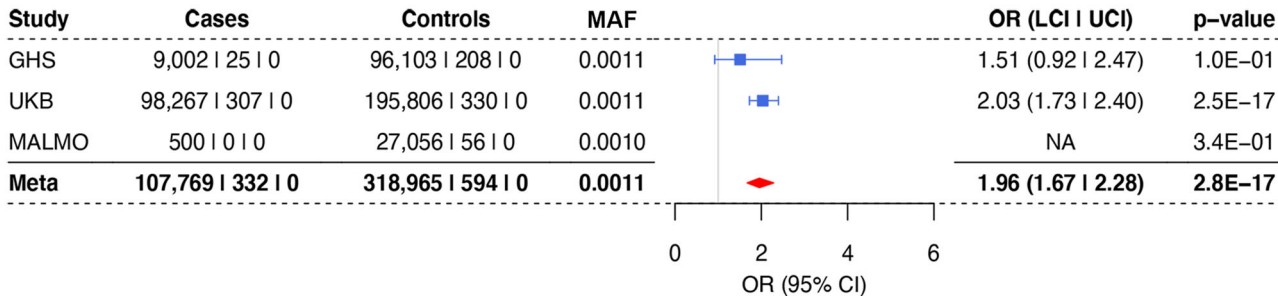

**Fig. 2 Heterozygous carriers of variants in *GJB2* and *SLC26A5* have increased risk for hearing loss.** Association of *GJB2* Gly12fs (**a**), *SLC26A5* Leu46Pro (**b**) and *SLC26A5* pLOF and strict deleterious missense, MAF ≤ 0.0001 (**c**) with hearing loss.

and Lys181fs ($P = 8 \times 10^{-8}$; Supplementary Data 7). In addition, we observed a common (MAF = 0.04) missense variant (Val504Met) within the last Kelch domain of *KLHDC7B* associated with increased risk for hearing loss (OR = 1.14, $P = 4 \times 10^{-26}$). This variant was also prioritized as the sole causal variant at this locus by FINEMAP (Supplementary Data 3).

We also identified rare coding associations in fascin actin-bundling protein 2 (*FSCN2*) and synaptojanin 2 (*SYNJ2*) with increased risk for hearing loss, both of which were recently observed in UKB[18] (Fig. 4). In *FSCN2*, an actin cross-linking protein[41–43], an aggregate of pLOF and deleterious missense variants was associated with hearing loss, with majority of the carriers in the burden having the His138Tyr variant. Conditioning on His138Tyr attenuated the significance of the burden test ($P = 3 \times 10^{-6}$ after conditioning on His138Tyr; Supplementary Data 7) but did not eliminate the signal, suggesting that other

variants in *FSCN2* may increase the risk for hearing loss. Mice homozygous for loss-of-function mutations in *Fscn2* present progressive hearing loss starting at 3 weeks and near deafness by 24 weeks due to degeneration of the outer hair cells in the cochlea[43]. In *SYNJ2*, we identified an association with a missense (Thr656Met)[18] variant that lies in the catalytic domain of this inositol polyphosphate 5-phosphatase[44]. Mice harboring homozygous mutations in *Synj2* that are predicted to reduce protein levels or the 5-phosphatase catalytic activity show progressive high-frequency hearing loss and a degeneration of hair cells that is most profound in the outer hair cells[45,46].

**GWAS/ExWAS supports a highly polygenic architecture of adult hearing loss.** Given that this is the largest sequencing study to date of adult hearing loss, and given the presence of Mendelian

## a. *KLHDC7B* aggregate of predicted loss-of-function variants (MAF ≤ 0.01)

| Study | Cases | Controls | MAF | OR (LCI ǀ UCI) | p-value |
|---|---|---|---|---|---|
| GHS | 8,998 ǀ 29 ǀ 0 | 96,149 ǀ 162 ǀ 0 | 0.0009 | 2.05 (1.32 ǀ 3.19) | 1.5E−03 |
| UKB | 98,123 ǀ 451 ǀ 0 | 195,687 ǀ 449 ǀ 0 | 0.0015 | 2.15 (1.88 ǀ 2.47) | 9.3E−28 |
| SINAI | 313 ǀ 1 ǀ 0 | 10,007 ǀ 11 ǀ 0 | 0.0006 | 14.26 (0.33 ǀ 619.33) | 1.7E−01 |
| MALMO | 500 ǀ 0 ǀ 0 | 27,092 ǀ 20 ǀ 0 | 0.0004 | NA | 5.7E−01 |
| **Meta** | **107,934 ǀ 481 ǀ 0** | **328,935 ǀ 642 ǀ 0** | **0.0013** | **2.14 (1.88 ǀ 2.45)** | **5.2E−30** |

## b. *KLHDC7B* (Lys181fs)

| Study | Cases | Controls | MAF | OR (LCI ǀ UCI) | p-value |
|---|---|---|---|---|---|
| GHS | 9,018 ǀ 8 ǀ 0 | 96,274 ǀ 34 ǀ 0 | 0.0002 | 2.90 (1.19 ǀ 7.08) | 1.9E−02 |
| UKB | 98,450 ǀ 118 ǀ 0 | 196,002 ǀ 121 ǀ 0 | 0.0004 | 2.14 (1.64 ǀ 2.78) | 1.9E−08 |
| **Meta** | **107,468 ǀ 126 ǀ 0** | **292,276 ǀ 155 ǀ 0** | **0.0004** | **2.19 (1.70 ǀ 2.82)** | **1.4E−09** |

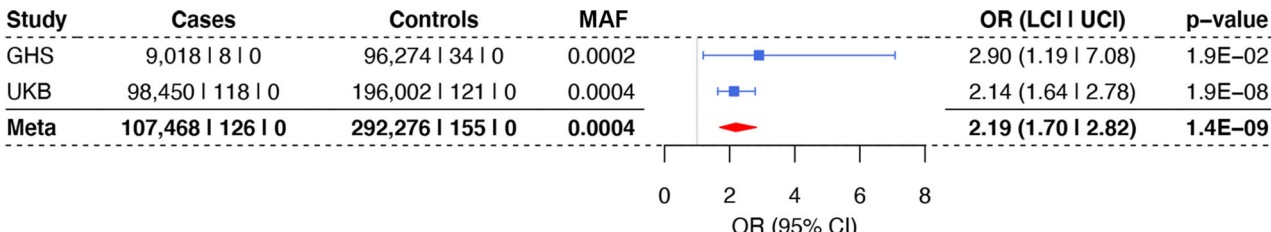

## c. *KLHDC7B* (Gly302fs)

| Study | Cases | Controls | MAF | OR (LCI ǀ UCI) | p-value |
|---|---|---|---|---|---|
| GHS | 9,013 ǀ 12 ǀ 0 | 96,259 ǀ 52 ǀ 0 | 0.0003 | 2.69 (1.32 ǀ 5.51) | 6.6E−03 |
| UKB | 98,377 ǀ 195 ǀ 0 | 195,956 ǀ 178 ǀ 0 | 0.0006 | 2.34 (1.89 ǀ 2.90) | 6.9E−15 |
| SINAI | 313 ǀ 1 ǀ 0 | 10,013 ǀ 9 ǀ 0 | 0.0005 | 49.78 (0.64 ǀ 3,887.84) | 7.9E−02 |
| **Meta** | **107,703 ǀ 208 ǀ 0** | **302,228 ǀ 239 ǀ 0** | **0.0005** | **2.39 (1.94 ǀ 2.93)** | **9.5E−17** |

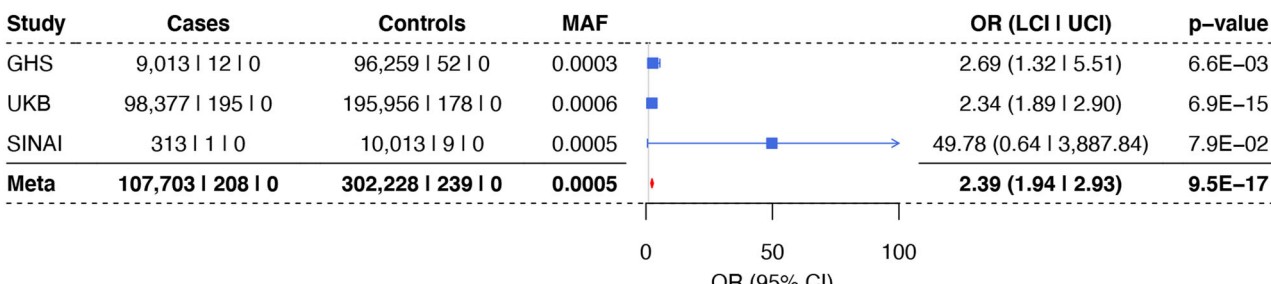

**Fig. 3 Association of rare variants in *KLHDC7B* with increased risk for hearing loss in meta-analysis. a** Association of an aggregate of predicted loss-of-function variants (MAF ≤ 0.01) in *KLHDC7B* with risk for hearing loss. **b, c** Two variants were the predominant contributors to the *KLHDC7B* loss-of-function gene burden aggregate, Lys181fs (**b**) and Gly302fs (**c**).

hearing loss genes among our common and rare-variant associations, we sought to explore the distribution of effect sizes across allele frequencies (Fig. 5). Hearing loss-associated variants span the frequency spectrum and, while we observe a few rare variants of large effect (e.g. *COCH* Cys542Phe and *MYO6* His246Arg), we do not observe any common variants of large effect. We further estimated phenotypic variance explained by the genetic data, or heritability $h^2_{Tot}$, using LD score regression (LDSC) partitioning into functional categories and stratifying by minor allele frequency[47,48]. We estimated a total heritability ($h^2_{Tot}$) of 0.089, with contributions from common variation ($h^2_{CV}$, MAF > 0.05) and low-frequency variation ($h^2_{LFV}$, 0.001 < MAF ≤ 0.05) of 0.074 and 0.015, respectively (Supplementary Fig. 3, Supplementary Data 11). These results indicate that, while the bulk of SNP heritability is derived from common variation, low-frequency variation contributes 16.8% of total SNP heritability.

## Discussion

We performed combined GWAS and ExWAS using exome-sequencing data and identified 53 independent associations, of which 15 have not previously been associated with hearing loss, to the best of our knowledge. The 15 associations included a missense lead variant in *TMPRSS3*, a known cause of Mendelian hearing loss, adding to the tally of Mendelian deafness genes (*EYA4, CDH23, TRIOBP*) showing common coding-variant associations with hearing loss in adult humans. We also identified coding variant/gene burden associations with 15 genes through exome sequencing. We estimated that low-frequency variation contributes a non-negligible portion (16.8%) of SNP heritability for adult hearing loss, consistent with a highly polygenic genetic architecture with rare, low-frequency and common genetic variation for adult hearing loss, and where variants of large effect would be subject to purifying selection.

### a. *FSCN2 aggregate of pLOF and missense (MAF ≤ 0.01)*

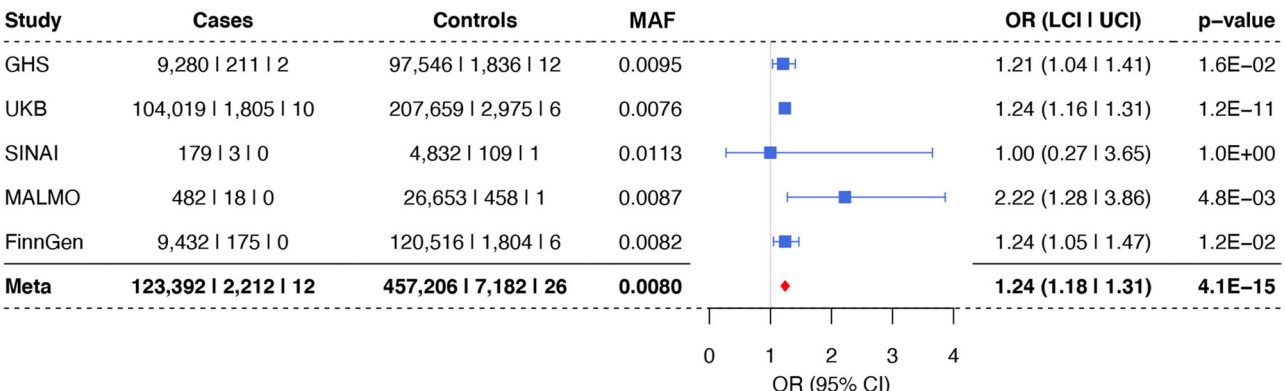

| Study | Cases | Controls | MAF | | OR (LCI \| UCI) | p–value |
|---|---|---|---|---|---|---|
| GHS | 8,446 \| 579 \| 2 | 90,863 \| 5,435 \| 13 | 0.0287 | | 1.15 (1.05 \| 1.27) | 2.6E–03 |
| UKB | 93,239 \| 5,321 \| 14 | 186,674 \| 9,446 \| 16 | 0.0252 | | 1.14 (1.10 \| 1.18) | 4.8E–13 |
| SINAI | 299 \| 15 \| 0 | 9,436 \| 579 \| 3 | 0.0290 | | 0.96 (0.57 \| 1.63) | 8.9E–01 |
| MALMO | 466 \| 34 \| 0 | 25,745 \| 1,363 \| 4 | 0.0254 | | 1.43 (0.99 \| 2.05) | 5.7E–02 |
| **Meta** | **102,450 \| 5,949 \| 16** | **312,718 \| 16,823 \| 36** | **0.0261** | | **1.14 (1.11 \| 1.18)** | **1.9E–15** |

### b. *FSCN2 (His138Tyr)*

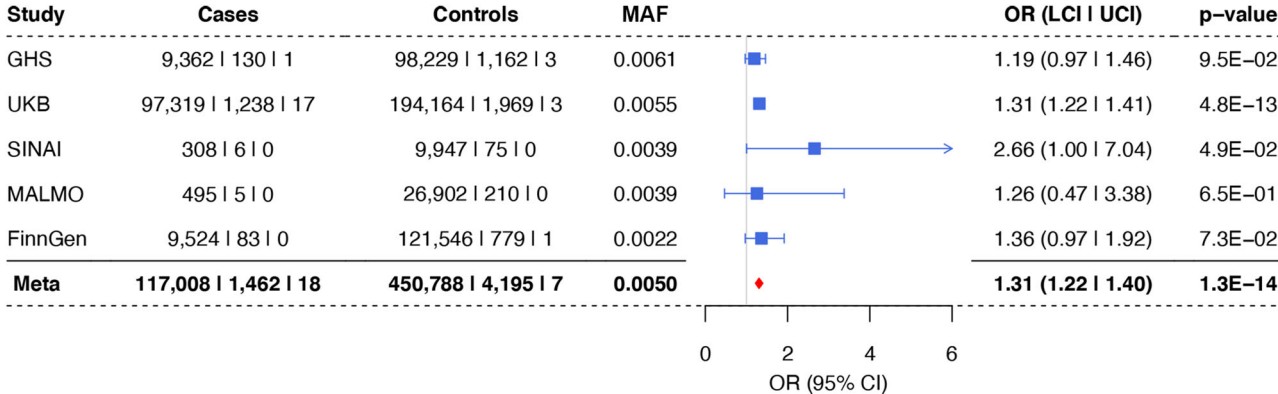

| Study | Cases | Controls | MAF | | OR (LCI \| UCI) | p–value |
|---|---|---|---|---|---|---|
| GHS | 9,280 \| 211 \| 2 | 97,546 \| 1,836 \| 12 | 0.0095 | | 1.21 (1.04 \| 1.41) | 1.6E–02 |
| UKB | 104,019 \| 1,805 \| 10 | 207,659 \| 2,975 \| 6 | 0.0076 | | 1.24 (1.16 \| 1.31) | 1.2E–11 |
| SINAI | 179 \| 3 \| 0 | 4,832 \| 109 \| 1 | 0.0113 | | 1.00 (0.27 \| 3.65) | 1.0E+00 |
| MALMO | 482 \| 18 \| 0 | 26,653 \| 458 \| 1 | 0.0087 | | 2.22 (1.28 \| 3.86) | 4.8E–03 |
| FinnGen | 9,432 \| 175 \| 0 | 120,516 \| 1,804 \| 6 | 0.0082 | | 1.24 (1.05 \| 1.47) | 1.2E–02 |
| **Meta** | **123,392 \| 2,212 \| 12** | **457,206 \| 7,182 \| 26** | **0.0080** | | **1.24 (1.18 \| 1.31)** | **4.1E–15** |

### c. *SYNJ2 (Thr656Met)*

| Study | Cases | Controls | MAF | | OR (LCI \| UCI) | p–value |
|---|---|---|---|---|---|---|
| GHS | 9,362 \| 130 \| 1 | 98,229 \| 1,162 \| 3 | 0.0061 | | 1.19 (0.97 \| 1.46) | 9.5E–02 |
| UKB | 97,319 \| 1,238 \| 17 | 194,164 \| 1,969 \| 3 | 0.0055 | | 1.31 (1.22 \| 1.41) | 4.8E–13 |
| SINAI | 308 \| 6 \| 0 | 9,947 \| 75 \| 0 | 0.0039 | | 2.66 (1.00 \| 7.04) | 4.9E–02 |
| MALMO | 495 \| 5 \| 0 | 26,902 \| 210 \| 0 | 0.0039 | | 1.26 (0.47 \| 3.38) | 6.5E–01 |
| FinnGen | 9,524 \| 83 \| 0 | 121,546 \| 779 \| 1 | 0.0022 | | 1.36 (0.97 \| 1.92) | 7.3E–02 |
| **Meta** | **117,008 \| 1,462 \| 18** | **450,788 \| 4,195 \| 7** | **0.0050** | | **1.31 (1.22 \| 1.40)** | **1.3E–14** |

**Fig. 4 Association of human hearing loss with genes previously implicated in hearing loss in mice. a** pLOF and missense (MAF ≤ 0.01) burden association in *FSCN2*. **b** The His138Tyr variant is the major contributor to the burden. **c** *SYNJ2* (Thr656Met) association with increased risk for hearing loss.

Notably, the majority of genes implicated by rare-variant and burden associations are already known to cause Mendelian forms of hearing loss; however, the odds ratios for these associations are wide-ranging. At the lower end of the effect spectrum are single-variant associations in *GJB2* and *SLC26A5*, with ORs close to typical for GWAS findings (1.2-1.3), followed by *COL11A2* (2.2-7) and, at the high end of the distribution, *COCH* and *MYO6* (31-81). *COCH* Cys542Phe and *MYO6* His246Arg are known pathogenic variants, and consistent with this, we observe almost all carriers presenting hearing loss. The two control carriers for each variant (Supplementary Data 6) could be explained by imperfect ascertainment of the phenotype or, as these mutations cause late-onset and progressive hearing loss, could also reflect a difference in expressivity. While some dilution of the effect sizes may be expected when working with self-reported as opposed to objective measures of hearing loss, the broad range in ORs that we observe suggests that for several of these Mendelian hearing loss genes we are identifying carriers with incompletely penetrant, risk-increasing variants. These results are consistent with the hypothesis that there is a continuum between Mendelian and common forms of hearing loss with the same genes harboring mutations causal for the former, and risk-increasing for the latter. Furthermore, compared to epidemiological risk factors for common hearing loss, including noise exposure (OR ~ 1.5–3[49]), odds

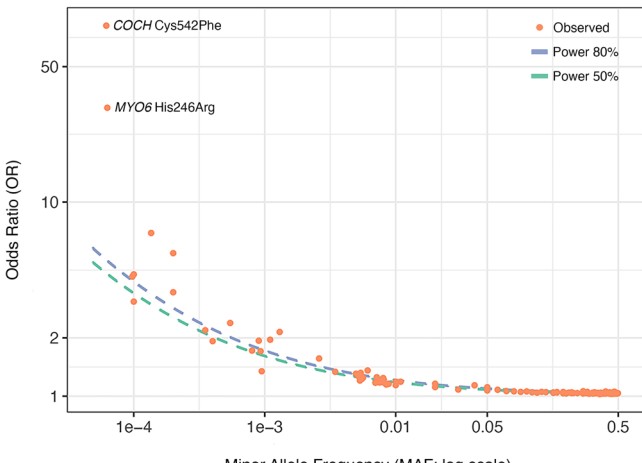

**Fig. 5 Effect size and allele frequency for variants associated with adult hearing loss.** Plotted are odds ratio estimates (on log scale) and minor allele frequencies of genome-wide significant variants and gene burdens (Supplementary Data 2, 6 and 7). 50% and 80% power curves for the present study are plotted as dotted lines. Notably, almost all points (particularly low-frequency and common variants, MAF > 0.0005) lie very close to the dotted power curves.

ratios of low-frequency and rare genetic factors may be large enough to provide mechanistic insights and to have implications for precision medicine.

Ultimately, to understand the biology of hearing loss and develop testable hypotheses from our association results, we need to identify the genes driving the observed associations at each locus. For common variants, we note the high resolution of FINEMAP to prioritize causal variants in several loci, including pinpointing single variants in four genes, three of which are related to hearing loss (*CDH23*), hearing function (*CTBP2*), or have significant rare variant associations with hearing loss (*KLHDC7B*). While FINEMAP helps prioritize variants, the gene that is impacted by those variants is not necessarily obvious, so we looked for colocalization with GTEx eQTL data and identified 19 genes whose expression may influence hearing loss risk (Supplementary Data 4). Analysis of single-cell expression data from mouse ears showed that 13 of the 19 genes are expressed in ear tissues. Of note, among these are *LMO7* and *SPTBN1* (BetaII-spectrin). Both genes code for components of the cuticular plate in the hair cells of the ear and knockout mice for either gene develop hearing loss[50,51]. *LMO7* colocalizes with eQTLs for the gene in GTEx skeletal muscle and *SPTBN1* in thyroid tissue, such that decreased expression of these genes is correlated with increased risk for hearing loss in humans.

One of the most compelling ways in which human genetics can establish a role for a gene in disease is when a diverse set of rare nonsynonymous or loss-of-function variants in the same gene show consistent association with disease. Coding variant and gene burden associations that we identified included associations with known pathogenic (e.g. in *MYO6* and *COCH*) and uncharacterized variants (*COL11A2* Phe80Ser), in genes that can cause autosomal dominant Mendelian hearing loss. We also identified variants and genes that cause recessive hearing loss (*GJB2* Gly12fs and *SLC26A5* pLOF+strict deleterious missense mutations), associated with increased risk (OR~1.2–1.3) for hearing loss in heterozygous carriers. Missense mutations in *GJB2* can cause dominant, syndromic hearing loss[52] but loss-of-function mutations, including Gly12fs, cause recessive non-syndromic hearing

loss[53–55]. Consistent with our findings, studies have detected hearing deficiency at high frequencies and a possibly more prominent effect in female adult heterozygous carriers of Gly12fs[55–57]. Mutations in *SLC26A5* cause recessive hearing loss in humans[58] and mice homozygous for *Slc26a5* null mutations develop hearing loss as early as 5–7 weeks of age[59]. *Slc26a5* heterozygous null mice have a hearing deficiency that is intermediate between the controls and knockouts. While we cannot rule out the possibility of compound heterozygosity with variation in regulatory/promoter regions in heterozygous carriers, our results offer the possibility that heterozygous carriers of loss-of-function variants in *GJB2* and *SLC26A5* may also have increased risk for adult-onset hearing loss.

Three of the 15 coding and burden associations were in genes that have not previously been implicated in Mendelian hearing loss in humans: *KLHDC7B*, *FSCN2*, and *SYNJ2*. In *KLHDC7B*, we confirmed a previously reported association of the common Val504Met and a rare frameshift variant with increased risk for hearing loss[18,20], and identified associations of increased risk with a burden of additional pLOF variants. While the biological effect of the missense is difficult to interpret, the association of putative LOFs with increased risk for hearing loss suggests that loss of KLHDC7B function is deleterious for hearing function. Based on the smaller effect of Val504Met (OR = 1.14) compared to the pLOF burden (OR = 2.14), we would hypothesize that this more common missense (MAF = 0.04) is a hypomorph. *KLHDC7B* (Kelch-like domain containing 7B) is a relatively understudied gene; it has been characterized as a 594-amino-acid protein containing a Kelch domain that is hypermethylated and upregulated in breast cancer cell lines and may influence cell proliferation in MCF-7 cells lines[60,61]. In the mouse ear, RNAseq expression profiling[62] and our real-time PCR data (Supplementary Fig. 4) show *Klhdc7B* expression in the cochlea with enrichment in the outer hair cells[63]. Consistent with the hypothesis from our genetic findings that loss of KLHDC7B function increases risk for hearing loss, initial characterization of *Klhdc7b* null mice by IMPC showed that homozygous carriers have hearing loss[28]. Mouse models *Fscn2* and *Synj2* also develop hearing loss[43,45,46]. Based on our association results, it would also be interesting to test heterozygous null *Klhdc7b*, *Fscn2*, and *Synj2* animals for increased susceptibility to hearing loss with age or environmental insults such as noise exposure.

We recognize that our study has several limitations. Our phenotype includes (in UKB) self-reported hearing loss among adults, which is likely to be a heterogeneous mix of early-onset, late-onset, age-related, as well as hearing loss due to environmental insults. In general, greater phenotype precision, including environmental exposure measures, should help future genetic analyses of adult hearing loss. We note that our colocalization analyses utilized GTEx eQTL data across tissues not including ear expression data. Given the high degree of sharing of genetic regulation of expression across tissues[64,65], the results are likely to point to the causal genes in many instances. eQTL data for ear cell types should help with the interpretation of genetic analyses of adult hearing loss.

In summary, this work contributes to connecting the two ends of the genetic architecture of hearing loss by detecting a common signal in genes known to cause hearing loss in Mendelian fashion and by detecting an additive signal (i.e. increased risk in heterozygous carriers) in genes known to cause autosomal recessive hearing loss. This latter finding also connects young- and adult-onset hearing loss in a single phenotypic spectrum with complex genetic underpinnings, including contributions from rare and common variation.

## Methods

**Participating cohorts and phenotype data.** We performed meta-analysis for hearing loss on a total of 125,749 cases and 469,497 controls of European ancestry from the following cohorts: United Kingdom Biobank (UKB)[66], the MyCode Community Health Initiative cohort from Geisinger Health System (GHS)[67], the Mount Sinai BioMe cohort (SINAI) (https://icahn.mssm.edu/research/ipm/programs/biome-biobank/pioneering), the Malmö Diet and Cancer study (MALMO)[68], and FinnGen R3 (https://www.finngen.fi/). Hearing loss case-control status in GHS, MALMO, and SINAI was defined using ICD-10 code diagnoses from the EHR. Cases were individuals with ICD10 H90.3-H90.8, H91.1or H91.9 diagnoses. Controls were individuals who did not meet the case criteria and did not have a diagnosis for ICD-10 Q16 (congenital malformations of ear causing hearing impairment) or ICD-10 H93.1 (tinnitus). In UKB the phenotype was defined using ICD-10 codes as above or by self-reported hearing loss or complete deafness on the touchscreen questionnaire (Field IDs: 2247 and 2257). Controls in UKB were individuals who did not have an ICD-10 diagnosis for hearing loss or tinnitus and did not self-report hearing loss, deafness or tinnitus (Field IDs: 4803 and non-cancer illness code 1597). The FinnGen analysis used was finngen_r3_H8_CONSENHEARINGLOSS, conductive or sensorineural hearing loss defined by ICD10 codes H90[.0-8], versus controls excluding other ear disorders (H91-H95). Further details are given in the Supplementary Information.

**Ethical approval and informed consent.** All participants provided informed consent, and studies were approved by the individual Institutional Review Boards (IRBs) at the respective institutions. UK Biobank has approval from the North West Multi-Centre Research Ethics Committee (MREC; ref: 11/NW/0382), which covers the UK. It also sought the approval in England and Wales from the Patient Information Advisory Group (PIAG) for gaining access to information that would allow it to invite people to participate. The DiscovEHR study was approved by the Geisinger Health System Institutional Review Board. The BioMe Biobank is an ongoing research biorepository approved by the Icahn School of Medicine at Mount Sinai's IRB. The Ethical Committee at Lund University approved the Malmo Diet and Cancer Study (LU 51-90). The Finngen Biobank was approved by the Coordinating Ethics Committee of the Helsinki and Uusimaa Hospital District.

**Genetic data and association analyses.** High-coverage whole-exome sequencing was performed at the Regeneron Genetics Center as previously described[69,70]. For SINAI and MALMO, DNA from participants was genotyped on the Global Screening Array (GSA), and for GHS genotyping was done on either the Illumina OmniExpress Exome (OMNI) or GSA array; the datasets (stratified by array for GHS) were imputed to the TOPMed (GHS) or HRC (MALMO and SINAI) reference panels using the University of Michigan Imputation Server (https://imputationserver.sph.umich.edu/index.html) or the TOPMed Imputation Server (https://imputation.biodatacatalyst.nhlbi.nih.gov/). Additional details are given in the Supplementary Information. We tested for association with hearing loss genetic variants or their gene burdens using REGENIE v1.0.43[71]. Analyses were adjusted for age, age[2], sex, an age-by-sex interaction term, experimental batch-related covariates, and genetic principal components. Cohort-specific statistical analysis details are provided in the Supplementary Information. Results across cohorts were pooled using inverse-variance weighted meta-analysis.

**Fine mapping and follow-on genetic analyses.** We defined genome-wide significant loci in our analysis by linkage disequilibrium ($r^2 > 0.1$) with lead variants. We defined previously associated loci by their index variants reported in previous hearing loss GWAS[13–17,19,20,72–75], and excluded 1 Mb regions surrounding them in the identification of previously unreported, to the best of our knowledge, loci in our analysis. LD score (LDSC) regression[76] was used to assess inflation (LDSC intercept) accounting for polygenic signal. Power calculations determined genotype relative risks (GRRs) providing 80 and 50 percent power given specified risk allele frequencies (RAFs, from $10^{-6}$ to 1) and the numbers of cases and controls in our meta-analysis. Fine mapping analyses included forward stepwise conditional analyses carried out in every locus with GCTA-COJO using a UK Biobank subsample LD reference panel, with independent associations determined using a joint $P$-value threshold of $1 \times 10^{-5}$ and $r^2$ cutoff of 0.9, and FINEMAP[34] Bayesian causal variant inference, using LD from available individual level data.

For genes whose cis regions overlapped genome-wide significant hearing loss loci, coloc2[35] was used to assess evidence for colocalization between our hearing loss GWAS and GTEx (release v8) cis-eQTL data derived from 48 tissues (https://www.gtexportal.org/home/), using GWAS and eQTL summary statistics for all common (MAF ≥ 0.01) variants within each gene's cis-region. Genes with posterior probability of colocalization H4 ≥ 0.5 were determined as having evidence for colocalization, and were visually inspected in eQTL+GWAS regional association plots[77].

Conditional analyses were performed for each cohort using REGENIE's Firth-corrected logistic regression and the resulting summary statistics were meta-analyzed as described above. Rare-variant association analyses conditional on the common variant signal were carried out for four loci with both common (MAF ≥ 0.01) and single rare variant (MAF < 0.01) genome-wide significant signals, by including as covariates the dosages of variants identified in fine mapping

analyses. Burden analyses conditional on rare variants were carried out for five genes with significant single rare variant and burden associations.

Assessment of heterozygous effects used association analyses excluding homozygotes as well as individuals carrying pairs of exome sequenced variants (MAF < 0.02 and MAC > 1) that were called as compound heterozygous mutations (CHMs) or potential CHMs (i.e. unknown phase). CHMs were called by SHAPEIT4 (https://github.com/odelaneau/shapeit4) phasing of merged genotype and exome data with scaffolds based on inferred close relatives[78,79].

Heritability derived from variants partitioned into seven functional categories (coding-synonymous, coding-nonsynonymous, 5-prime-UTR, 3-prime-UTR, splice site, intronic, intergenic) with each category further stratified into a low frequency (0.001 < MAF ≤ 0.05) and common (MAF > 0.05) minor allele frequency bin as in stratified LD score regression was estimated using LD score regression (LDSC) of hearing loss association statistics on LD scores. LD scores were generated from a reference panel of $N = 10,000$ random UK Biobank European-ancestry samples' merged imputed and exome data.

**Single-cell RNA sequencing and analysis.** All protocols were approved by the Institutional Animal Care and Use Committee in accordance with the Regeneron's Institutional Animal Care and Use Committee (IACUC). Cochlea and utricles from C57BL/6 mice at post-natal day 7 were micro-dissected and dissociated. Suspensions of 200 cells/μL were subjected to Chromium Single Cell (10x Genomics) library preparation and were sequenced on Illumina NextSeq 500. Cell Ranger Single-Cell Software Suite (10x Genomics, v2.0.0) was used to perform sample de-multiplexing, alignment to MM10 Genome assembly with UCSC gene models, filtering, and UMI counting. PCA, UMAP and clustering analyses used Seurat V3.2 (https://github.com/satijalab/seurat)[80]. Cluster marker genes as well as canonical cell type-specific genes were used to manually label the cell type for each cluster.

**Statistics and reproducibility.** For genome-wide association meta-analysis, the statistical threshold of $P < 5 \times 10^{-8}$ was considered statistically significant. For simplicity and reproducibility, meta-analysis was performed to combine statistical results across cohorts rather than a multi-stage design or requiring nominal significance in multiple cohorts; sample sizes are reported above and in Supplementary Data 1. In conditional analyses of common variants within loci, $P < 5 \times 10^{-8}$ was considered as a threshold for reporting independent associations within loci. Rare-variant associations were reported as independent of common variants in the same locus when they were $P < 5 \times 10^{-3}$, as determined by Bonferroni correction for 10 conditional tests performed. We describe burden results conditional on individual variants within the aggregate burden counts regardless of significance, in order to determine if any variants were driving the burden signal. Further details are given above and in the Supplementary Note.

**Reporting summary.** Further information on research design is available in the Nature Research Reporting Summary linked to this article.

## Data availability

All whole-exome sequencing, genotyping chip, and imputed sequence for UKB described in this report are publicly available to registered researchers via the UK Biobank data access protocol. Additional information about registration for access to the data is available at http://www.ukbiobank.ac.uk/register-apply/. Further information about the whole-exome sequence is available at http://www.ukbiobank.ac.uk/wp-content/uploads/2019/03/Access_064-UK-Biobank-50k-Exome-Release-FAQ-v3.pdf Detailed information about the chip and imputed sequence is available at: http://www.ukbiobank.ac.uk/wp-content/uploads/2018/03/UKB-Genotyping-and-Imputation-Data-Release-FAQ-v3-2-1.pdf. Geisinger DiscovEHR, Malmo Diet and Cancer study and Mt. Sinai Biome exome-sequencing and genotyping data can be made available to qualified, academic, non-commercial researchers upon request via a Data Transfer Agreement with the respective institutions. Summary statistics for FinnGen r3 can be downloaded from https://www.finngen.fi/en/access_results. Regeneron materials (RNA sequencing data) described in this manuscript are available to qualified, academic, non-commercial researchers upon request through Regeneron portal (https://regeneron.envisionpharma.com/vt_regeneron/) after signing Material/Data Transfer Agreements with Regeneron. Regeneron may deny any request for RNASeq data made by or on behalf of a recipient outside of the academic community; for any use other than to replicate or extend the results described in this publication, including any commercial use or use sponsored by a commercial entity.

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

## Acknowledgements

The authors would like to thank everyone who made this work possible. In particular: the UK Biobank team, their funders, the dedicated professionals from the member institutions who contributed to and supported this work, and most especially the UK Biobank participants, without whom this research would not be possible. The exome sequencing was funded by the UK Biobank Exome Sequencing Consortium (i.e., Bristol Myers Squibb, Regeneron, Biogen, Takeda, Abbvie, Alnylam, AstraZeneca, and Pfizer). This research has been conducted using the UK Biobank Resource under application number 2604. We also want to acknowledge the participants and investigators of the FinnGen study. We thank the MyCode Community Health Initiative participants for taking part in the DiscovEHR collaboration.

## Author contributions

All authors contributed to critical review of the manuscript for important intellectual content, and final approval of submission of the manuscript for publication. Conceptualization: K.P., E.A.S., M.D., B.Z., A.B., G.C. Analysis: K.P., L.D., L.G., M.A.F., A.H.A., J.S., A.P., A.M., S.C., E.A.S., C.B., J.M. Single-cell sequencing data generation and analysis: S.M., Y.B., J.S. Mouse experiments: A.K. Project administration: E.C., M.J. Datasets: O.M. Supervision: M.A.F., G.A., E.A.S., S.B., B.Z., G.C., A.B. Writing – original draft: K.P., L.D., E.A.S., G.C. All authors contributed to securing funding, study design and oversight. All authors reviewed the final version of the manuscript. C.B., C.F., A.L., and J.D.O. performed and are responsible for sample genotyping. C.B, C.F., E.D.F., M.L., M.S.P., L.W., S.E.W., A.L., and J.D.O. performed and are responsible for exome sequencing. T.D.S., Z.G., A.L., and J.D.O. conceived and are responsible for laboratory automation. M.S.P., K.M., R.U., and J.D.O are responsible for sample tracking and the library information management system. X.B., A.H., O.K., A.M., S.O., R.P., T.P., A.R., W.S. and J.G.R. performed and are responsible for the compute logistics, analysis and infrastructure needed to produce exome and genotype data. G.E., M.O., M.N. and J.G.R. provided compute infrastructure development and operational support. S.B., S.K., and J.G.R. provide variant and gene annotations and their functional interpretation of variants. E.M., J.S., R.L., B.B., A.B., L.H., J.G.R. conceived and are responsible for creating, developing, and deploying analysis platforms and computational methods for analyzing genomic data. All authors contributed to the clinical informatics of the project. All authors contributed to the analysis of the project. All authors contributed to the management and coordination of all research activities, planning and execution. All authors contributed to the review process for the final version of the manuscript. All authors contributed towards the creation of the GHS-RGC DiscovEHR collaboration, helped frame research questions and contributed to the discussion and review of data and results, review and feedback on manuscript, and contributed to the management and coordination of discussions.

## Competing interests

The authors declare the following competing interests: Regeneron authors receive salary from and own options and/or stock of the company. Decibel authors receive salary and may own options and/or stock of the company. The remaining authors declare no competing interests.

## Additional information

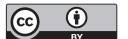

## GHS-REGN DiscovEHR collaboration

Lance J. Adams[5], Jackie Blank[5], Dale Bodian[5], Derek Boris[5], Adam Buchanan[5], David J. Carey[5], Ryan D. Colonie[5], F. Daniel Davis[5], Dustin N. Hartzel[5], Melissa Kelly[5], H. Lester Kirchner[5], Joseph B. Leader[5], David H. Ledbetter[5], J. Neil Manus[5], Christa L. Martin[5], Raghu P. Metpally[5], Michelle Meyer[5], Tooraj Mirshahi[5], Matthew Oetjens[5], Thomas Nate Person[5], Christopher Still[5], Natasha Strande[5], Amy Sturm[5], Jen Wagner[5] & Marc Williams[5]

[5]Geisinger, Danville, PA 17821, USA.

## Regeneron Genetics Center

**RGC Management and Leadership Team** Goncalo R. Abecasis [1], Aris Baras[1], Michael Cantor[1], Giovanni Coppola[1], Andrew Deubler[1], Aris Economides[1], Luca A. Lotta[1], John D. Overton[1], Jeffrey G. Reid[1], Alan Shuldiner[1], Katia Karalis[1] & Katherine Siminovitch[1]

**Sequencing and Lab Operations** Christina Beechert[1], Caitlin Forsythe[1], Erin D. Fuller[1], Zhenhua Gu[1], Michael Lattari[1], Alexander Lopez[1], John D. Overton[1], Thomas D. Schleicher[1], Maria Sotiropoulos Padilla[1], Louis Widom[1], Sarah E. Wolf[1], Manasi Pradhan[1], Kia Manoochehri[1] & Ricardo H. Ulloa[1]

**Genome Informatics** Xiaodong Bai[1], Suganthi Balasubramanian[1], Boris Boutkov[1], Gisu Eom[1], Lukas Habegger[1], Alicia Hawes[1], Shareef Khalid[1], Olga Krasheninina[1], Rouel Lanche[1], Adam J. Mansfield[1], Evan K. Maxwell[1], Mona Nafde[1], Sean O'Keeffe[1], Max Orelus[1], Razvan Panea[1], Tommy Polanco[1], Ayesha Rasool[1], Jeffrey G. Reid[1], William Salerno[1] & Jeffrey C. Staples[1]

**Clinical Informatics** Nilanjana Banerjee[1], Michael Cantor[1], Dadong Li[1], Deepika Sharma[1] & Ashish Yadav[1]

**Translational and Analytical Genetics** Alessandro Di Gioia[1] & Sahar Gelfman[1]

**Research Program Management** Esteban Chen[1], Marcus B. Jones[1], Jason Mighty[1], Michelle G. LeBlanc[1] & Lyndon J. Mitnaul[1]

## Decibel-REGN collaboration

**Collaboration Core Team** Joe Burns[6], Giovanni Coppola[1], Meghan C. Drummond[2], Aris Economides[1,2], David Frendewey[2], Scott Gallagher[6], John Lee[6], John Keilty[6], Christos Kyratsous[2], Lynn Macdonald[2], Adam T. Palermo[6], Kavita Praveen[1], Leah Sabin[2], Jonathon Whitton[6] & Brian Zambrowicz[2]

**Program Management & Alliance Management** Sarah Deng[2], Geoff Horwitz[6], Alejandra K. King[2] & Jung H. Sung[2]

[6]Decibel Therapeutics, Boston, MA, USA.

