## [Peer Review File · Communications Biology]

Reviewers' comments:

Reviewer #1 (Remarks to the Author):

The Praveen et al manuscript conducts a generally well done and straightforward GWAS of hearing loss, the largest such one to date, and I enjoyed reading. I appreciate that some extra work was done with tissue expression. A few issues.

The results mention that COJO was run, but the Methods suggest a conditional analysis was run using REGENIE and individual level data. Which is it?

While there may be some difficulty with COJO assumptions, depending on what was done for the conditional analysis, why weren't individuals of non-European ancestry included, e.g., from the UKB?

How did REGENIE heritability estimates compare to LD score estimates?

The data availability section does not include the single cell mouse expression data?

Reviewer #2 (Remarks to the Author):

This manuscript details the largest WGS study for adult onset hearing loss to date. It provides very important information for the field regarding common and rare variants associated with hearing loss. Much of the work to date focused on common variation. In this article, the authors do an excellent job with sound statistical analyses of demonstrating a significant amount of the genetic load associated with this problem. I think it makes a tremendous contribution to the field.

Reviewer #3 (Remarks to the Author):

This is a very clearly explained and well executed project which contributes to our understanding of the genetic predisposition of age related hearing impairment (ARHI). The authors have studied common and rare genetic variants in available cohorts having hearing phenotyping to identify associations, many of which have been reported before but some of which are novel. The quality of the writing is high and the finding of 15 novel associations through common and rare variant analysis makes an important contribution to the genetics of hearing.

Major comments

1. In common with many common complex trait GWAS meta-analyses the samples are highly heterogeneous in size and dominated by the UK Biobank contribution. It would be clearer for the reader to understand up front which of the existing reports have used UK Biobank and which haven't, in the Introduction. Re-arranging the introduction like this would set the scene for this manuscript's contribution, particularly to move from simple GWAS associations to the novel use of UK Biobank exomes and the resultant hearing associations.

2. There are multiple sub-studies reported in this manuscript and while the methods describes them clearly there is opacity around the level of significance considered for each statistical test and how this has been adjusted for multiple testing. The methods include this information for the fine mapping but could similar information be provided for each other step please?

Minor comments

1. There is an imbalance in the sex ratio of the mice selected with only one male mouse. Please reassure readers that there is no apparent difference between the sexes - would it be better to omit the single male altogether?
2. Figure 1 - the legend includes A and B but these do not show up on (my version) of the figure. Please amend

line 109 - "While the majority"

line 201 - pLOF please define where first used

Population-scale analysis of common and rare genetic variation
associated with hearing loss in adults

Kavita Praveen¹, Lee Dobbyn¹, Lauren Gurski¹, Ariane H. Ayer¹, Jeffrey Staples¹, Shawn
Mishra², Yu Bai², Alexandra Kaufman², Arden Moscati¹, Christian Benner¹, Esteban Chen¹,
Siying Chen¹, Alexander Popov¹, Janell Smith², GHS-REGN DiscovEHR collaboration*,
Regeneron Genetics Center*, Decibel-REGN collaboration*, Olle Melander^{3,4}, Marcus Jones¹,
Jonathan Marchini¹, Suganthi Balasubramanian¹, Brian Zambrowicz², Meghan Drummond², Aris
Baras¹, Goncalo R. Abecasis¹, Manuel A. Ferreira¹, Eli A. Stahl¹, Giovanni Coppola¹

From:

¹Regeneron Genetics Center, Tarrytown, NY 10591, USA.

²Regeneron Pharmaceuticals, Inc., Tarrytown, NY 10591, USA.

³Lund University, Department of Clinical Sciences Malmö, Malmö, Sweden.

⁴Skåne University Hospital, Department of Emergency and Internal Medicine, Malmö, Sweden.

*A complete list of contributing authors to the Regeneron Genetics Center, GHS-RGC DiscovEHR

Collaboration and the Decibel-REGN Collaborations is provided in the Supplementary Appendix.

Correspondence to giovanni.coppola@regeneron.com.

**ABSTRACT**

[revised manuscript text omitted]

Recently, Ivarsdottir et al¹⁸ published association results with hearing loss on ~50K
Icelandic individuals with whole-genome sequence and ~50K individuals from UKB with exome
sequence data, with imputation of larger samples into these variant sets. We have now expanded
the rare variant analysis to exome-sequences from ~295K individuals in UKB and ~143K
individuals from three other datasets. Here, we report findings from genome- and exome-wide
association meta analyses with a total of 125,749 cases and 469,497 controls. Our analyses have
identified 15 novel susceptibility loci and 15 rare variant associations that provide novel
[revised manuscript text omitted]

**URLs**

International Mouse Phenotyping Consortium: <https://www.mousephenotype.org/>.

gEAR portal: <https://umgear.org/>
GTEEx: <https://www.gtexportal.org/home/>
[referred to in METHODS]
Mount Sinai BioMe: <https://icahn.mssm.edu/research/ipm/programs/biome-biobank/pioneering>
FinnGen: <https://www.finnngen.fi/>
University of Michigan Imputation Server: <https://imputationserver.sph.umich.edu/index.html>
TOPMed Imputation Server: <https://imputation.biodatacatalyst.nhlbi.nih.gov/>
MakeScaffold: <https://github.com/odelaneau/makeScaffold>
SHAPEIT4: <https://github.com/odelaneau/shapeit4>
GTEEx v8 eQTL data: <https://www.gtexportal.org/>
Seurat V3.2: <https://github.com/satijalab/seurat>

**ACKNOWLEDGEMENTS**

The authors would like to thank everyone who made this work possible. In particular: the UK
Biobank team, their funders, the dedicated professionals from the member institutions who
contributed to and supported this work, and most especially the UK Biobank participants, without
whom this research would not be possible. The exome sequencing was funded by the UK Biobank
Exome Sequencing Consortium (i.e., Bristol Myers Squibb, Regeneron, Biogen, Takeda, Abbvie,
Alnylam, AstraZeneca and Pfizer). This research has been conducted using the UK Biobank
Resource under application number 2604. We also want to acknowledge the participants and
investigators of the FinnGen study. We thank the MyCode Community Health Initiative
participants for taking part in the DiscovEHR collaboration.

[revised manuscript text omitted]

REVIEWERS' COMMENTS:

Reviewer #3 (Remarks to the Author):

The authors have addressed the reviewers' comments satisfactorily.